# A neurophysiological basis for aperiodic EEG and the background spectral trend

Niklas Brake [1,2], Flavie Duc[3], Alexander Rokos [3], Francis Arseneau[3], Shiva Shahiri [4], Anmar Khadra [2] ✉ & Gilles Plourde [3] ✉

Electroencephalograms (EEGs) display a mixture of rhythmic and broadband fluctuations, the latter manifesting as an apparent 1/f spectral trend. While network oscillations are known to generate rhythmic EEG, the neural basis of broadband EEG remains unexplained. Here, we use biophysical modelling to show that aperiodic neural activity can generate detectable scalp potentials and shape broadband EEG features, but that these aperiodic signals do not significantly perturb brain rhythm quantification. Further model analysis demonstrated that rhythmic EEG signals are profoundly corrupted by shifts in synapse properties. To examine this scenario, we recorded EEGs of human subjects being administered propofol, a general anesthetic and GABA receptor agonist. Drug administration caused broadband EEG changes that quantitatively matched propofol's known effects on GABA receptors. We used our model to correct for these confounding broadband changes, which revealed that delta power, uniquely, increased within seconds of individuals losing consciousness. Altogether, this work details how EEG signals are shaped by neurophysiological factors other than brain rhythms and elucidates how these signals can undermine traditional EEG interpretation.

Electroencephalograms (EEGs) display a mixture of periodic and aperiodic fluctuations. Almost a century of research has established that periodic EEG signals are generated by synchronous neural oscillations[1–4]. In contrast, aperiodic EEG signals remain relatively poorly understood. Whereas periodic EEG signals produce peaks in power spectra, the aperiodic component manifests as the background spectral trend that decays with apparent $1/f^\beta$ behaviour[5–8]. Differences in the spectral exponent, $\beta$, have been correlated with aging, cognitive performance, neurological disorders, anesthesia, and sleep[9–14]. In addition to being a useful biomarker, it has been proposed that the EEG spectral trend may change independently of neural oscillations and that spectral detrending is necessary to accurately quantify differences in brain rhythms[14]. Deciphering the neurophysiological basis of aperiodic EEG is thus necessary for correctly interpreting EEG biomarkers and for improving algorithms that quantify brain rhythms.

There exist two main hypotheses for how the EEG spectral trend is generated by the brain. The synaptic timescale hypothesis predicts that the EEG spectral trend is a natural consequence of exponentially decaying synaptic currents, and that consequently asynchronous network activity will produce a spectrum with a $1/(1 + \tau^2 f^2)$ or "Lorentzian" trend[7,15,16]. The second hypothesis is based on the theory of self-organized criticality which posits that the propagation of action potentials throughout neural networks produces so-called "avalanches" of activity with magnitudes following a 1/f distribution[17–19]. It has been hypothesized that such avalanche dynamics in turn generate a 1/f trend in EEG[8,20,21]. These theories are different in two important respects. Firstly, the avalanche hypothesis implies that the trend in EEG spectra is informative about the dynamics of the brain, while the synaptic timescales hypothesis is agnostic. Secondly, the two theories suggest different shapes for the spectral trend and therefore propose distinct methods for detrending EEG spectra[14].

[1]Quantiative Life Sciences PhD Program, McGill University, Montreal, Canada. [2]Department of Physiology, McGill University, Montreal, Canada. [3]Department of Anesthesia, McGill University, Montreal, Canada. [4]School of Nursing, McGill University, Montreal, Canada. ✉e-mail: anmar.khadra@mcgill.ca; gilles.plourde@mcgill.ca

Despite these hypotheses, the concept of aperiodic EEG itself remains controversial. Some argue that the apparent trend in EEG spectra is an epiphenomenon caused by slower brain rhythms recruiting larger populations of neurons[22,23]. According to this viewpoint, the EEG spectrum does not require detrending and the spectral exponent is a conflated measure of various changes in brain rhythms.

Three questions therefore remain open: (1) can EEG signals reflect arrhythmic neural activity? (2) if so, how do these signals shape EEG spectra? (3) do EEG spectra need to be detrended, and if so, what is the most physiologically meaningful method of detrending? To investigate these questions, we combined numerical forward modelling of scalp potentials with biophysical calculations of single-neuron dipoles[24–26]. With this approach, we simulated biophysically realistic EEG signals generated by networks exhibiting a range of dynamics. These simulations revealed several mechanisms, besides brain rhythms, that affect EEG signals and together shape the spectral trend. To test model predictions, we recorded EEG of humans receiving an infusion of the drug propofol, a general anesthetic that targets GABA receptors and slows the decay time of inhibitory synaptic currents[27–30]. These experiments identified specific EEG changes during propofol administration that were expected to contaminate brain rhythm estimates. Using our modelling insights, we corrected for these sources of contamination and reevaluated known EEG signatures of losing consciousness. Overall, this study develops a biophysically grounded theory describing the neural basis of aperiodic scalp potentials and provides practical conclusions for the spectral analysis of EEG data.

## Results

### EEG cannot reflect asynchronous neural activity

To understand the neurophysiology that underlies the EEG spectral trend, we began by investigating the properties of EEG generation at the single-neuron level. These properties are informative because they shape the ensemble EEG regardless of coherence or neural synchrony

(See Definitions and theoretical framework in Methods). To examine these properties, we performed simulations of biophysically and morphologically detailed neuron models (Fig. 1a). To start, we did not assume any dynamics of presynaptic neurons and therefore modelled synaptic input with independent Poisson spike trains (Fig. 1a), an assumption that we relax later. From these currents, a single-neuron dipole was computed[26] (Fig. 1b). The neuron was placed at a random location in the cortex and the single-neuron EEG signal was calculated using the New York head model[24] (Fig. 1c). This entire procedure was repeated many times with various representative neuron morphologies (Table S1) and with the neurons placed at various locations in the cortex. The average spectrum, which we will refer to as a unitary spectrum, is the expected EEG spectrum generated by a single average cortical neuron (Fig. 1d).

The unitary spectrum exhibited two important features. First, even with random synaptic input, the unitary spectrum displayed a trend. This trend could be described as the sum of two Lorentzian functions

$$S(f) = \frac{A_1}{1 + (2\pi\tau_1 f)^2} + \frac{A_2}{1 + (2\pi\tau_2 f)^2}, \tag{1}$$

as predicted by simple linear models of EEG generation[7,16] (Fig. 1d). The slower ($\tau_1$) and faster ($\tau_2$) timescales were governed by the deactivation kinetics of GABA receptors (GABARs) and AMPA receptors (AMPARs), respectively (Fig. 1e, f). These simulations validate previous predictions that the relative contribution of excitatory and inhibitory currents to the EEG signal fundamentally affects the spectral trend[16].

Second, the unitary spectrum reflects the amplitude with which an average neuron contributes to the EEG signal. To investigate this idea in more detail, we examined the effect of varying all the model parameters within physiologically reasonable ranges (Fig. 1g). Measuring the average power of the resulting single-neuron EEGs revealed

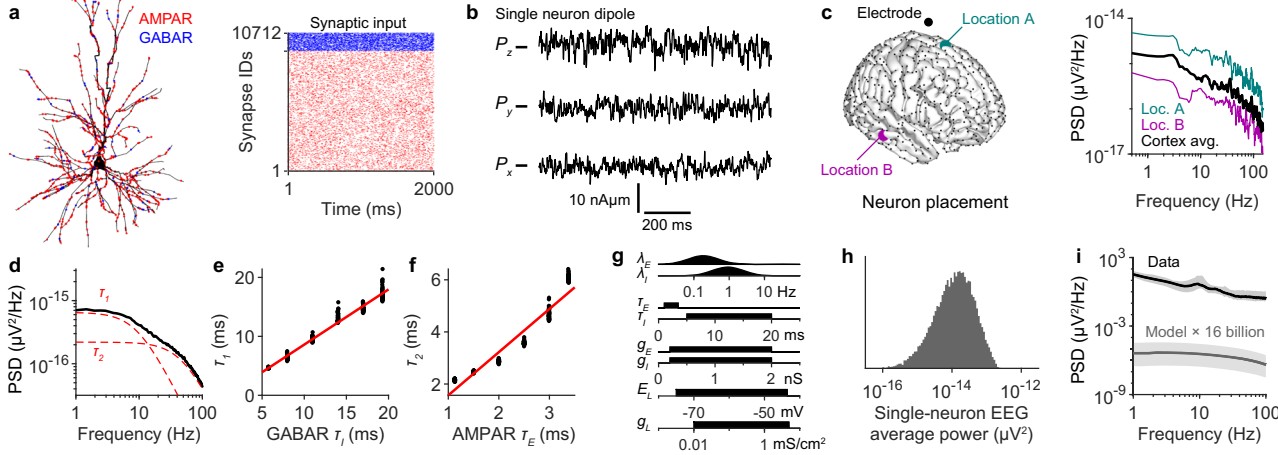

**Fig. 1 | EEG cannot reflect asynchronous neural activity. a** Example morphology of a layer 2/3 pyramidal neuron. Inputs at AMPAR (red) and GABAR (blue) synapses were simulated with Poissonian spike trains, shown in raster plot. Only 1000 synapses shown for clarity. Neuron morphology adapted from Budd, J. M. L. et al. Neocortical axon arbors trade-off material and conduction delay conservation. PLoS Comput. Biol. 6, e1000711 (2010). **b** The x, y, and z components of the single-neuron dipole vector, calculated from (**a**). **c** Left: single-neuron EEG signals were simulated at the marked electrode location, with the neuron located at various source locations, e.g., location A and B. Right: the location-averaged EEG spectrum (black) computed by averaging over the source locations shown in black dots on the brain template. Loc.=location; Avg.=average. **d** Location-averaged spectrum generated from 1000 simulations of 11 representative neuron morphologies (Table S1). The spectrum was fit by Eq. 1 and the two Lorentzian components are shown in dashed red lines. **e** The unitary spectrum was calculated while varying the

deactivation kinetics of GABARs ($\tau_I$) and the parameter $\tau_1$ was estimated. Red line has a slope of 1. **f** Same as **e**, but showing $\tau_2$ as a function of the deactivation kinetics of AMPARs ($\tau_E$). **g** Sampling distributions for model parameters. $\lambda_E$, $\tau_E$, and $g_E$ represent the average input rate, the deactivation time constant, and maximal conductance for AMPAR synapses. $\lambda_I$, $\tau_I$, and $g_I$ represent the same parameters for GABAR synapses. $E_L$ and $g_L$ represent the reversal potential and conductance of the passive membrane leak current. **h** Distribution of single-neuron EEG power (location-averaged as in **c**) based on 20,000 simulations with parameters sampled from the distributions shown in (**g**) and morphologies sampled as described in Table S1. **i** Black: Median EEG spectrum across 14 subjects, with error bands indicating minimum and maximum spectral density. Grey: the predicted EEG spectrum of 16 billion uncorrelated neurons receiving Poissonian synaptic input (grey), with parameter values sampled from the distributions in (**g**). Error bands reflect 5–95% quantile range.

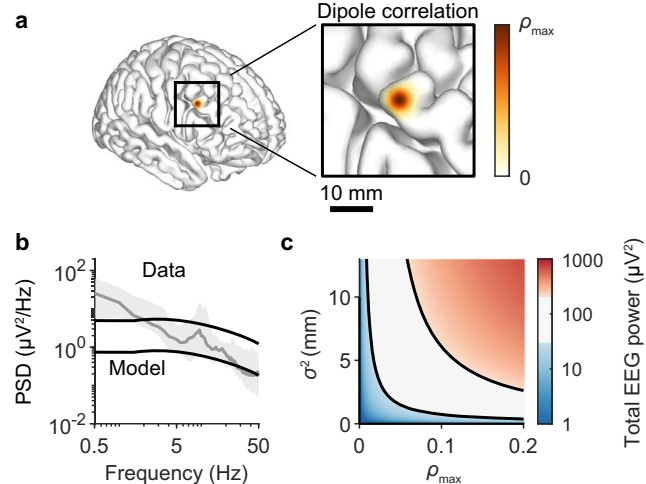

**Fig. 2 | Detectable EEG signals require only weak, locally synchronized dipoles.**
**a** Schematic displaying the coupling kernel used to correlate single-neuron dipoles.
The kernel is a Gaussian function with peak of $\rho_{max}$ and variance $\sigma^2$. **b** Median EEG
spectrum across 14 human subjects (grey; same error bands as Fig. 1i), and simu-
lated unitary spectrum (black; same as Fig. 1d) scaled by arbitrary amounts to
determine a lower and upper bound of the spectral trend amplitude. **c** Heatmap
of the total simulated EEG power produced by 16 billion neurons with dipoles coupled
with the kernel in (**a**), and parameterized with various values of $\rho_{max}$ and $\sigma^2$. The
black lines are level curves representing the lower and upper bounds for the
spectral trend magnitude obtained in (**b**).

a range of possible amplitudes for the contribution of individual
neurons to the EEG signal (Fig. 1h). These simulations show that no
physiologically plausible parameters could allow 16 billion uncorre-
lated neurons to generate detectable EEG signals (Fig. 1i). More
importantly, these calculations quantify how far off a completely
asynchronous cortex is from generating detectable EEG signals. These
calculations account for the EEG strength contributed by synaptic
current amplitudes, neuron geometry, average firing rate of neurons,
the number of neurons, the conductivity of various tissues, and the
geometry of the cortex. Therefore, this result indicates that neural
dynamics are responsible for the approximate four orders of magni-
tude difference between the simulated asynchronous spectrum and
real EEG recordings.

### Detectable EEG requires only weak, locally synchronized dipoles

If the amplitude of EEG signals precludes asynchronous activity from
shaping EEG spectra, what types of arrhythmic activity could influence
EEG signals? To begin addressing this question, we first quantified in
general how much dipole correlation is required to generate detect-
able EEG signals. To do so, we imposed a simplified spatial organiza-
tion on a cortical template whereby neighbouring neurons produced
correlated dipoles. Specifically, the dipoles of neurons separated by $d$
mm were correlated with a coefficient of $\rho(d) = \rho_{max} \exp(-d^2/\sigma^2)$,
where $\rho_{max}$ is the maximal dipole correlation and $\sigma^2$ is the spatial scale
over which correlations decline (Fig. 2a). Based on this correlation
scheme, the geometry of the cortex, average densities of neurons, and
the single-neuron EEG amplitudes from the previous section, we esti-
mated the power of the ensemble EEG signal that would be produced
with different values of $\rho_{max}$ and $\sigma^2$ (see Methods). To capture the EEG
spectral trend, we estimated that broadband EEG signals need to have
a total power of between 50 and 200 $\mu V^2$ (Fig. 2b). Thus for neural
activity to generate detectable EEG signals, this activity needs to be
capable of driving single-neuron dipoles that are correlated up to a
value ($\rho_{max}$) of 0.06–0.12 (Fig. 1c), assuming a liberal range for $\sigma^2$ of
5–13 mm (Fig. 1c; see Methods). While the existence of EEG rhythms
prove that neural oscillations can generate the requisite dipole

synchrony, it remains to be determined if and how such a degree of
dipole coherence may be achieved by arrhythmic neural activity.

### Synapse topology is sufficient for dipole correlations

To investigate the ability of neural activity to generate coherent
dipoles, we simulated dyads of neurons and investigated the level of
correlation achievable between the two single-neuron dipoles. In our
simulations, we noticed that if a single synapse was activated, the
orientation of the resulting dipole could be accurately predicted by the
orientation of the synapse relative to the soma (Fig. 3a, b). This result
held across all neuron morphologies investigated (Fig. S1). From this
observation, we devised a minimal model of dipole coherence. To
generate synaptic input, we projected the synapses of two neurons
onto a sphere and correlated the inputs of synapses with close angular
distances (Fig. 3c). By changing the maximal correlation between
synapses, dipole correlation between neurons of various morpholo-
gies could be tuned continuously between 0 and -0.3 (Fig. 3d, e).
Shuffling the location of the synapses abolished any dipole correlation
(Fig. 3d). These results demonstrate that to generate detectable EEG
signals, it is necessary and sufficient for synaptic input to exhibit
temporal and spatial correlation. These conditions do not by them-
selves preclude any specific type of neural dynamics. Indeed this
minimal model relied on independently sampled spikes at every time
point and thus generated EEG signals with no temporal autocorrelation
and a spectrum with the same shape as the asynchronous unitary
spectrum (Fig. 3f). Therefore, the plausibility of aperiodic EEG signals
depends solely on the ability for a network of neurons to generate
arrhythmic activity with the requisite levels of spatiotemporal
correlation.

### Subcritical networks can generate aperiodic EEG signals

We next investigated network models that could generate detectable,
aperiodic EEG signals. To determine whether network activity could
produce coherent dipoles, we utilized the results from the previous
section (Fig. 3). After simulating a presynaptic neuron population, we
used the UMAP algorithm[31] to embed the population onto a sphere in
such a way that minimized the distance between presynaptic neurons
with correlated spiking activity (Fig. 4a). These presynaptic neurons
were then projected onto the dendrites of the postsynaptic dyad
(Fig. 4a). By construction, synapses with higher correlations should
have smaller angular distances, thus optimizing for dipole coherence.
Effectively, this procedure tests whether it is geometrically possible for
a network to generate sufficiently coherent synaptic input for EEG
generation.

To understand the mechanisms of aperiodic EEG generation, we
attempted to construct the simplest network that could generate
dipole coherence. Our previous results demonstrate that randomly
connected networks cannot generate coherent dipoles, because they
cannot produce spatially correlated activity. We therefore continued
on to use the simplest neural network that exhibits spatial topology,
namely, a spatial network. To construct the network, each neuron was
embedded in a plane and connected preferentially to nearby neurons
(Fig. 4b). For simplicity, we modelled individual neurons as binary
nodes, i.e., each neuron was either spiking or quiescent. Spikes pro-
pagated along network connections, causing subsequent neurons to
fire with some probability (Fig. 4b). This simple model falls within the
category of a branching network, because the dynamics are governed
by a single parameter called the branching number, denoted by $m$,
which reflects the average number of spikes successfully elicited
across the network when a single neuron fires.

Our simulations revealed that the maximal achievable dipole
correlation increased with the network's branching number (Fig. 4c, d).
A purely asynchronous network ($m = 0$) was unsurprisingly incapable
of generating correlations among single-neuron dipoles (Fig. 4d). On
the other hand, a slightly subcritical network ($m = 0.98$) was capable of

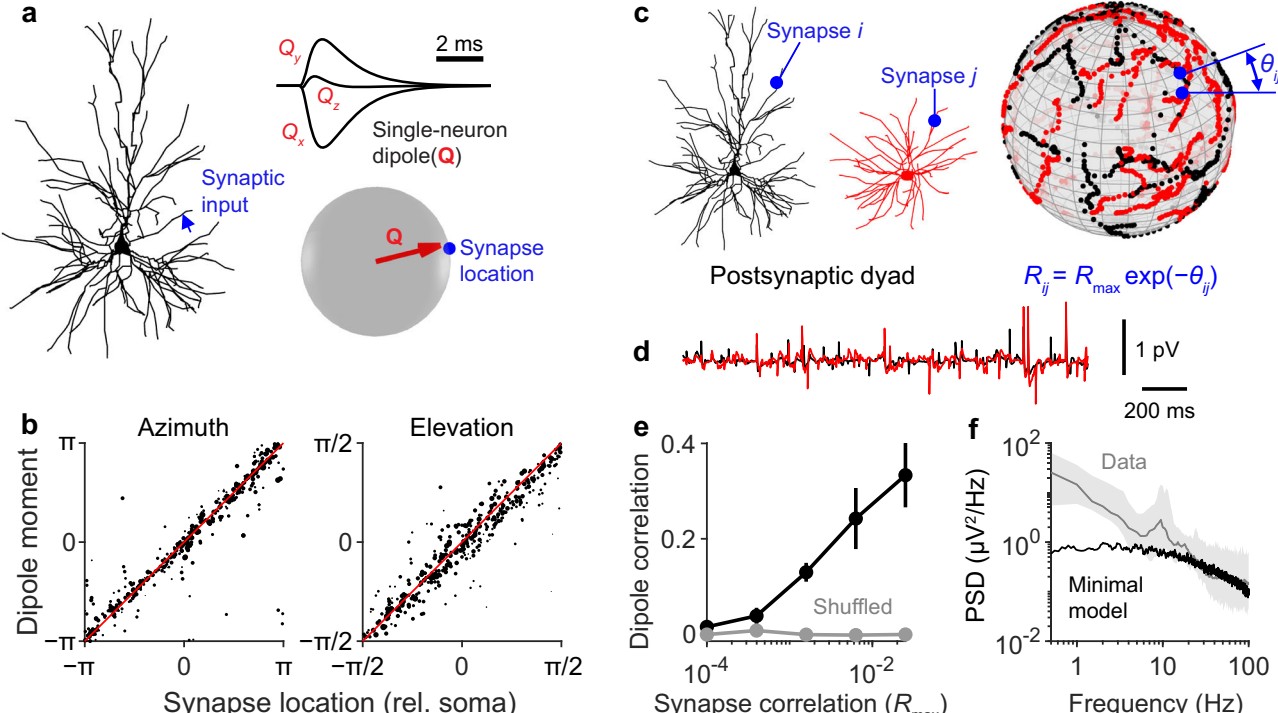

**Fig. 3 | A minimal model of dipole correlation captures broadband EEG amplitude but not low frequency power. a** A single synapse was activated at the location specified by the blue arrow, generating a response in the single-neuron dipole, **Q**. The dipole vector at the peak of the response is oriented towards the synapse location, defined in spherical coordinates with the soma as the origin. Neuron morphology adapted from Budd, J. M. L. et al. Neocortical axon arbors trade-off material and conduction delay conservation. PLoS Comput. Biol. 6, e1000711 (2010). **b** The simulation in **a** was repeated 600 times with different neuron morphologies and synapse locations. Plotting the dipole orientation at the peak of the response against the location of the stimulated synapse shows a strong, linear relationship. Some points not plotted for clarity. Dot size is proportional to peak dipole amplitude. rel.=relative. **c** Schematic showing the minimal model for dipole correlation. Synapses on each postsynaptic neuron in the dyad were projected onto a sphere. A correlation matrix among all synapses was then defined

such that synapses separated by an angle $\theta_{ij}$ on the sphere were correlated by $R_{max} \exp(-\theta_{ij})$. The left neuron's morphology is adapted from Budd, J. M. L. et al. Neocortical axon arbors trade-off material and conduction delay conservation. PLoS Comput. Biol. 6, e1000711 (2010). The illustration of the right neuron is adapted, with permission from SNCSC, from Mainen, Z. F. & Sejnowski, T. J. Influence of dendritic structure on firing pattern in model neocortical neurons. Nature 382, 363–366 (1996), Springer Nature. **d** Example of single-neuron EEG signals of the two neurons shown in (**c**). **e** Correlation between two single-neuron dipoles as a function of $R_{max}$. Vertical lines represent 95% confidence intervals of the mean ($n = 11$ different morphology pairs). When synapse locations are shuffled, dipoles are no longer correlated (grey). **f** The unitary spectrum of neurons receiving input calculated with the minimal model (black), scaled to compare the shape of the spectrum with median EEG spectrum of 14 subjects (grey line; same as Fig. 1i).

generating dipoles that were correlated by 0.18 ± 0.001 for an average pair of postsynaptic neurons (mean ± SE, n = 10,000; morphologies sampled as described in Table S1; biophysical parameters sampled as in Fig. 1g). This branching number is significant because previous work has found that networks with a branching number of $m = 0.98$ closely reproduced the in vivo dynamics of cortical spiking across many different species[32,33], making this a physiologically plausible parameter value. Placing synapses suboptimality led to proportionally lower dipole correlation, revealing that subcritical dynamics can still drive coherent dipoles with suboptimal synapse placement (Fig. 4e). Based on the amplitude of the computed unitary spectrum (Fig. 4f), we estimated that subcritical network activity can produce realistic EEG amplitudes with synapse optimality as low as ~25% (Fig. 4g; see Methods). As a reference, an optimality index of 50% means that strongly correlated synapses will on average lie on the same hemisphere of their respective neurons, a geometry reminiscent of the apical-basal compartmentalization of cortical pyramidal neurons.

Notably, stronger dipole correlations coincided with longer temporal autocorrelations in network activity (Fig. 4f), a phenomenon directly resulting from the causality of spike propagation (Fig. 4b). This means that EEG signals produced by propagating cortical spikes must have higher power at low frequencies if the signals are to be of detectable amplitudes. This result illustrates a fundamental constraint that may in part explain the 1/f scaling of EEG spectra at lower

frequencies (Fig. 4g). More broadly, these results demonstrate a biophysically feasible mechanisms that allows arrhythmic neural activity to generate EEG signals, and to consequently influence the broadband features of EEG spectra.

**Narrowband EEG changes could result from arrhythmic activity**
The above calculations show that arrhythmic neural activity can in theory generate broadband EEG signals, meaning that narrowband EEG power need not reflect brain rhythms. We therefore asked whether EEG signals that are produced by arrhythmic activity can confound traditional EEG interpretation, namely, that changes in bandpower reflect differences in neural oscillations. To address this question, we considered a scenario where the EEG signal is generated by two subpopulations of neurons: a population exhibiting synchronous oscillations, and a second population exhibiting subcritical dynamics ($m = 0.98$). If these population are completely independent of one another, their EEG contributions will by definition add together linearly. To investigate further, we simulated the scenario in which the two populations' postsynaptic signals are intermixed (Fig. 5a). One interpretation of this setup could be a cortical neuron that is receiving oscillatory input from, say, the thalamus, while also receiving subcritical input from the local cortical circuitry. Compared to neurons receiving entirely oscillatory input or entirely subcritical input, the neuron receiving mixed input produced a unitary EEG spectrum that

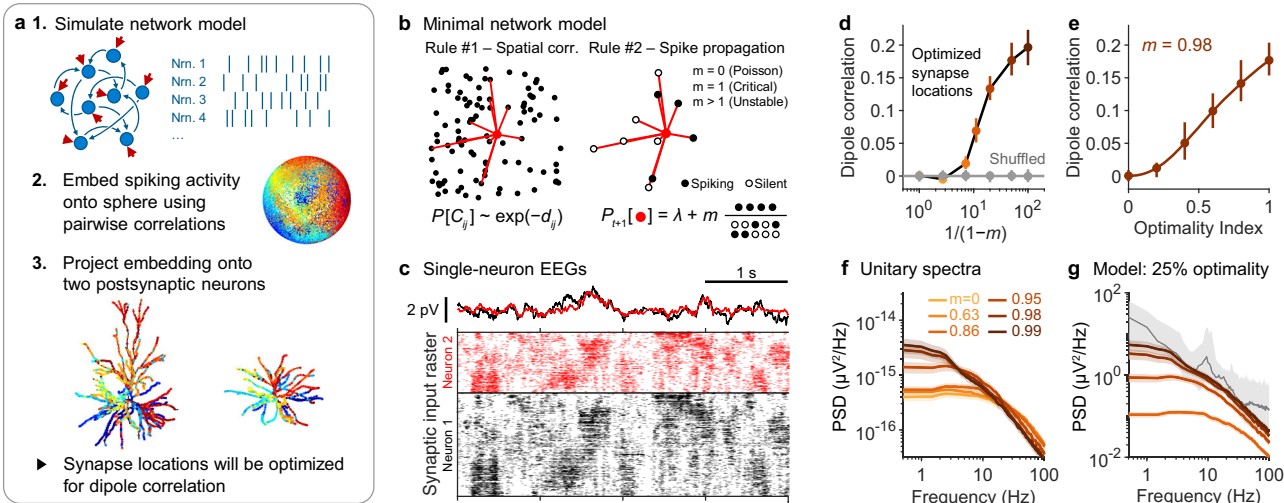

**Fig. 4 | Subcritical network dynamics can explain the amplitude and low frequency power of broadband EEG. a** Illustration of the algorithm for optimizing dipole coherence. Nrn. = neuron. The left neuron morphology is adapted from Budd, J. M. L. et al. Neocortical axon arbors trade-off material and conduction delay conservation. PLoS Comput. Biol. 6, e1000711 (2010). The illustration of the right neuron is adapted, with permission from SNCSC, from Mainen, Z. F. & Sejnowski, T. J. Influence of dendritic structure on firing pattern in model neocortical neurons. Nature 382, 363–366 (1996), Springer Nature. **b** Illustration of the rules used to produce spatiotemporal synchrony in a presynaptic network. Rule #1: network topology is enforced by making the probability of pairwise connections, $P[C_{ij}]$, decrease exponentially with distance, $d_{ij}$. Rule #2: the probability of a neuron firing, $P_{t+1}$, depends on a baseline firing rate of $\lambda = \lambda_0(1 - m)$ plus the average activity of its neighbours scaled by $m$. $\lambda_0$ is a predefined average firing rate; $m$ is the branching number of the network. corr.=correlation. **c** Examples of single-neuron EEG signals of two neurons receiving input from the network model with a branching number

$m = 0.98$, when synapse locations have been optimized as described in (**a**). Bottom: a raster plot of synaptic input for the two neurons. **d** Correlation between two single-neuron dipoles as a function of the branching number, $m$, of the presynaptic network. Dipole correlation increases with $m$ (coloured dots), but not when synapse locations are shuffled (grey). Median (dots) and 5– 95% quantile range (vertical lines) across $n = 10{,}000$ simulated dyads with randomly sampled morphologies (Table S1) and randomly sampled biophysical parameter values (as in Fig. 1g). **e** Dipole correlation can be tuned by placing synapses suboptimally (see Methods). Cubic interpolation between simulated optimality indices is shown with the solid line. Median (dots) and 5–95% quantile range (vertical lines) across $n = 500$ simulations. **f** Unitary spectra calculated for different branching numbers. Error bands reflect 95% confidence interval of the mean. **g** The unitary spectra from (**f**) scaled based on a synapse optimality index of 0.25 (colours same as in **f**). The value of $\rho_{max}$ was determined from **e** and the final scaling factor was determined as in Fig. 2. Grey: median EEG spectrum of 14 subjects (same as Fig. 1i).

exhibited a mixed phenotype (Fig. 5b). Increasing the strength of the oscillatory and/or aperiodic input altered the amplitude of the spectral peak and/or spectral trend (Fig. 5c–e). However, changes to the spectral trend did not multiplicatively scale the oscillatory peak amplitude. As a result, quantifying the amplitude of oscillatory peaks relative to the spectral trend produced incorrect interpretations (Fig. 5f–h): dividing the peak amplitude by the background trend erroneously suggested that the neural rhythm decreased when aperiodic activity became stronger (Fig. 5d, g), and severely underestimated the neural rhythm increase when aperiodic activity increased concomitantly (Fig. 5e, h). We therefore concluded that if a spectral peak is clearly discernible, then arrhythmic neural activity has a minimal confounding influence and detrending is unnecessary. Importantly, however, if no spectral peak is obvious, then there is no guarantee that changes in power result from differences in neural oscillations.

### Spectral slope is an inconsistent measure of EI balance
The above results focused on the role of neural dynamics in shaping the EEG spectrum. However, Fig. 1d–f shows that the kinetics of postsynaptic responses also impact the broadband properties of EEG spectra. To further investigate this mechanism shaping EEG spectra, we performed a full sensitivity analysis of the spectral slope with respect to the biophysical parameters in the single-neuron models. The unitary EEG spectrum was simulated with many different parameter values (Fig. 6a) and the overall slope of the spectrum was calculated[14] between 1 and 40 Hz (Fig. 6b). These simulations revealed that the spectral slope was consistently affected by four parameters: $\tau_I$, $g_E$, $g_I$, and $E_L$ (Fig. 6c). These results can be explained through Eq. 1. The parameter $\tau_I$ directly determines the slow timescale, $\tau_1$, of the unitary spectrum (Fig. 1d), whereas synaptic conductances, $g_E$ and $g_I$,

directly govern the contributions of inhibitory and excitatory synaptic currents, respectively. The reversal potential, $E_L$, of the leak current alters the spectral slope because when $E_L$ is more depolarized, GABA$_A$ receptors experience a higher driving force, thus amplifying the contributions of inhibitory currents.

Surprisingly, the analysis of the spectral slope revealed a strong interaction between $g_L$ and both $\lambda_E$ and $\lambda_I$ (Fig. 6c). When the leak conductance was low ($g_L$< 0.1 mS cm$^{-2}$), the spectral slope was found to be negatively correlated with the $\lambda_E$:$\lambda_I$ ratio (Fig. 6d, e; Fig. S2b), which contradicts the predictions of linear models of EEG generation[16]. The reason is that when the leak conductance was low, higher E:I ratios drove the average membrane potential close to the reversal potentials of GABA receptors (Fig. S2a). This caused neurons with low $g_L$ to experience vanishing GABAR driving forces and amplified AMPAR driving forces at high E:I values, consequently and counterintuitively making the spectral slope anticorrelated with the E:I ratio (Fig. 6e). However, when the leak conductance was high ($g_L$>1 mS cm$^{-2}$), the membrane potential fluctuated less and stayed within a relatively linear regime (Fig. S2a). Consequently the spectral slope was positively correlated with E:I ratio (Fig. 6e) as predicted by linear models of EEG generation[16]. A similar albeit weaker effect was observed in the relationship between the spectral slope and the $g_E$:$g_I$ ratio (Fig. S2c). Generally, we conclude that the overall spectral trend is significantly affected by biophysical parameters that alter postsynaptic responses, but changes in the slope value do not alone inform what parameters are changing.

### Changes to synaptic responses confound brain rhythm quantification
Differences in postsynaptic responses alter EEG spectra in a fundamentally different way to arrhythmic neural activity. Postsynaptic

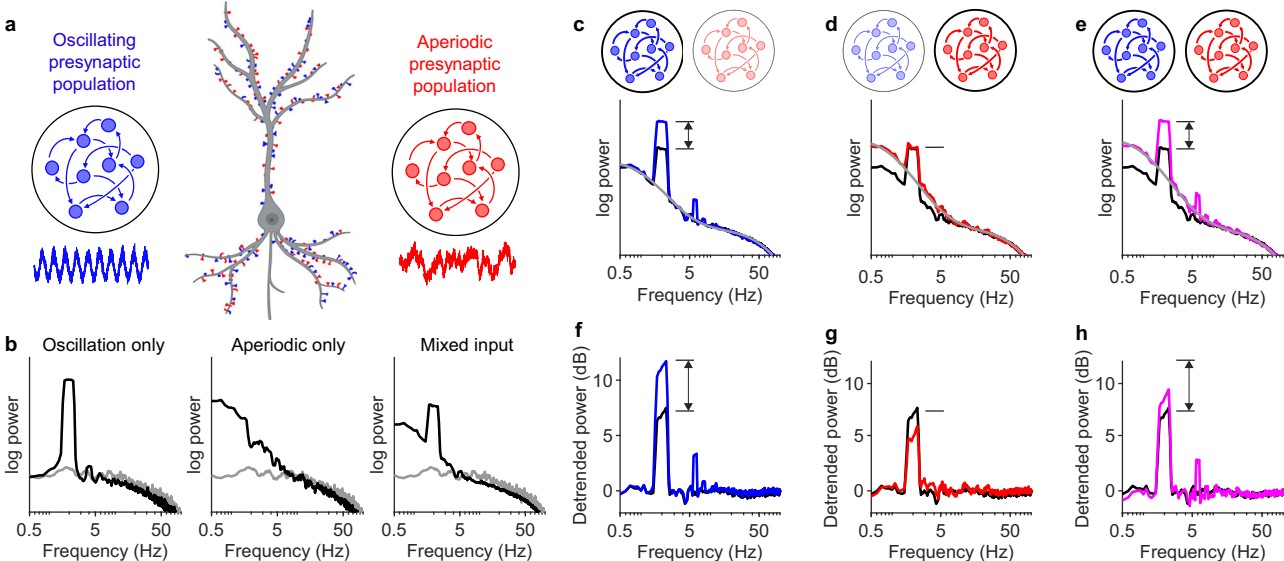

**Fig. 5 | Measuring oscillatory peak power relative to spectral trend may produce misleading results. a** Illustration of mixed input model. Half of the neuron's synapses received oscillatory rhythmic input (blue) and the other half received input from a subcritical network (red). The strengths of oscillatory and subcritical dynamics were adjusted by tuning two parameters, $\alpha_R$ and $\alpha_A$, respectively, which determined the degree to which synaptic inputs differed from homogenous Poisson processes. See Methods for details. Neuron illustration created with BioRender.com. **b** Unitary spectrum for neurons receiving Poisson input (grey), compared with the unitary spectra for neurons receiving input entirely from the oscillatory population (left; black), entirely from the subcritical population (middle; black), or mixed input as in (**a**) (right; black): $\alpha_R = \alpha_A = 0.1$. **c** Oscillatory input was strengthened by increasing $\alpha_R$ from 0.1 (black) to 0.5 (blue), with $\alpha_A$ fixed at 0.1. The spectra were fit using a FOOOF-like algorithm[14], except that here the aperiodic

component was modelled with Eq. 1 (solid grey line). **d** Aperiodic input was strengthened by increasing $\alpha_A$ from 0.1 (black) to 0.5 (red), with $\alpha_R$ fixed at 0.1. **e** Both oscillatory and aperiodic inputs were strengthened by increasing both $\alpha_R$ and $\alpha_A$ from 0.1 (black) to 0.5 (magenta). **f** Detrended spectrum of neurons receiving mixed input before (black) and after (blue) oscillation strength increased. The unitary spectra in (**c**) were divided by the solid grey line. **g** Detrended power before (black) and after (red) aperiodic strength increases. Notice that the detrended power at 2 Hz decreases, despite the strength of oscillations remaining the same. **h** Detrended power before (black) and after (magenta) both oscillation and aperiodic strength increases. Notice that the detrended power at 2 Hz does not increase as much as in (**f**), despite the oscillation increasing in strength by the same amount.

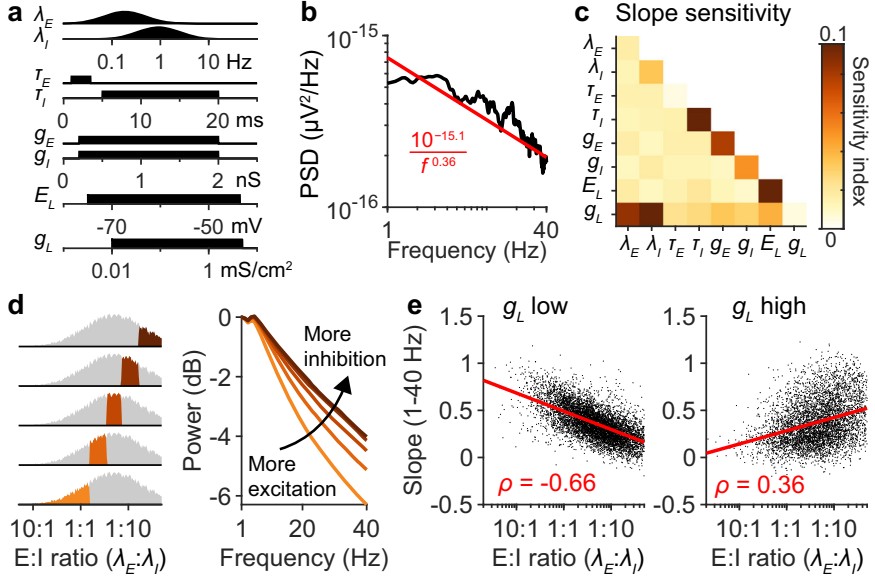

**Fig. 6 | Sensitivity of spectral slope to biophysical parameters governing postsynaptic responses. a** Sampling distributions for model parameters, the same as the ones used in Fig. 1g. **b** Example single-neuron EEG spectrum, fitted with the equation $10^\alpha / f^\beta$ between 1 and 40 Hz. **c** Sensitivity of the spectral slope, $\beta$, to model parameters, with first and second order interactions, calculated from 20,000

simulated spectra. **d** Simulated spectra were averaged depending on the ratio between $\lambda_E$ and $\lambda_I$. Lower E:I ratios correspond to more inhibition. **e** The spectral slope is plotted against the E:I ratio for each simulation. Left: simulation where the leak conductance was low ($g_L < 0.1$ mS cm$^{-2}$; $n = 7366$ simulations). Right: simulation where the leak conductance was high ($g_L > 1$ mS cm$^{-2}$; $n = 5184$ simulations).

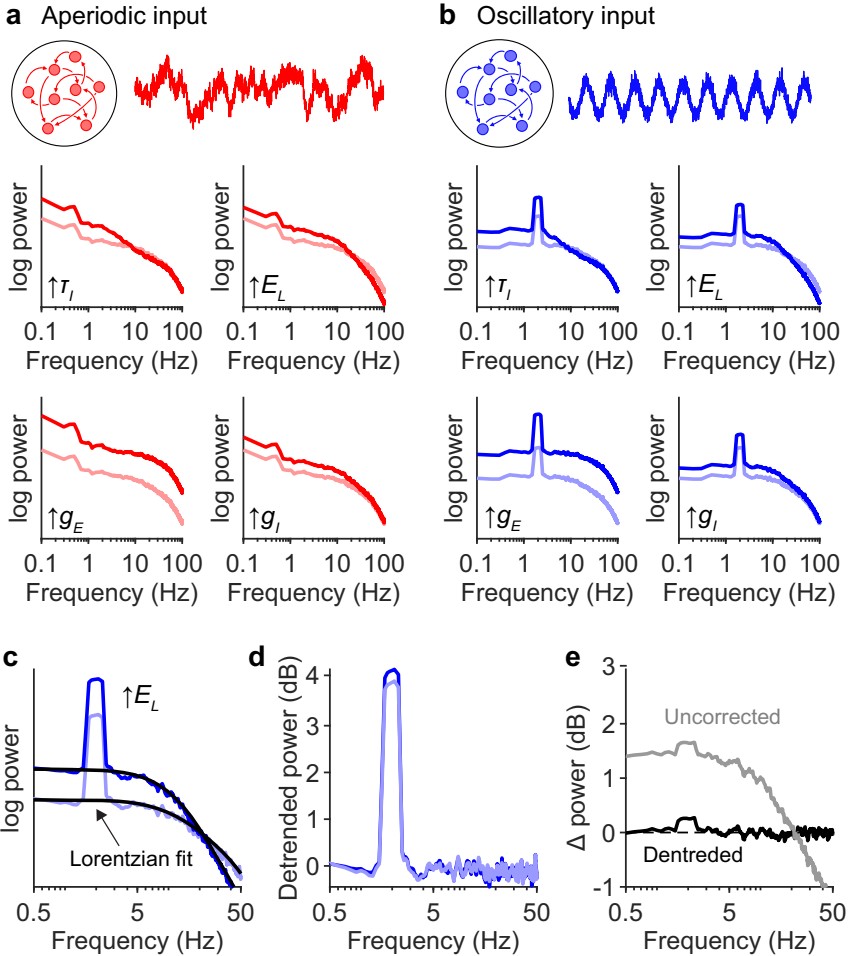

**Fig. 7 | Changes in postsynaptic mechanisms confound brain rhythm quantification. a** Unitary spectra for neurons receiving entirely subcritical input ($m = 0.98, \alpha_A = 0.1$; see Fig. 5a), before and after four parameter changes: subplot corresponding to ($\uparrow \tau_I$) shows the unitary spectrum when $\tau_I$ was increased from 10 ms (pink) to 30 ms (red); ($\uparrow E_L$) shows unitary spectrum when $E_L$ was increased from −60 mV (pink) to −45 mV (red); ($\uparrow g_E$) shows unitary spectrum when $g_E$ was increased from 0.7 nS (pink) to 1.4 nS (red); and ($\uparrow g_I$) shows unitary spectrum when $g_I$ was increased from 0.7 nS (pink) to 1.4 nS (red). Note the changes in the unitary spectra despite there being no changes in synaptic input dynamics. **b** Same as in (**a**), but for neurons receiving entirely oscillatory input ($\alpha_R = 0.1$; see Fig. 5a). In

Fig. S3, we show examples of different types of rhythmic input. **c** Example of fitted spectral trend. Here, the unitary spectrum of sinusoidal input is shown before and after increasing $E_L$. The spectra were fit using a FOOOF-like algorithm[14], except that here the trend was modelled using Eq. 1 (black lines). See other examples in Fig. S4. **d** Spectra from (**c**), detrended by dividing the spectra by their respective Lorentzian fits. See other examples in Fig. S4. **e** Change in EEG spectral density caused by increasing $E_L$. In grey is the raw change in the unitary spectra, while in black is the difference between the detrended spectra. Detrending the spectra with Eq. 1 has corrected for the effects of increasing $E_L$ and correctly indicates that there are no changes in neural dynamics.

kinetics effectively interact as a convolution with synaptic input and should therefore interact with oscillatory peaks in a multiplicative manner. Moreover, postsynaptic mechanisms should affect power generated by all types of neural dynamics equally. To test this, we systematically altered the biophysical parameters that govern postsynaptic currents and analyzed the unitary spectra generated by various types of synaptic inputs, including white noise (Fig. S3), subcritical dynamics (Fig. 7a), as well as three types of dynamics that exhibit spectral peaks: a recently proposed recurrent Ising model that exhibits co-existence of oscillations and avalanches[34] (Fig. S3), an underdamped second-order system driven by white noise (Fig. S3), and finally a simple sinusoidal rhythm (Fig. 7b). We investigated the effects of neurophysiological parameters that affect the spectral slope, as identified by our sensitivity analysis (Fig. 6c): GABAR kinetics ($\tau_I$), the leak current reversal potential ($E_L$), and the conductances of excitatory ($g_E$) and inhibitory ($g_I$) synapses. Changing these parameters altered the unitary spectra in distinct manners, but importantly had identical effects across the different types of the input dynamics (Fig. 7a, b; Fig. S3).

The spectral changes caused by altering biophysical parameters do not reflect differences in neural dynamics and therefore represent confounds for EEG analysis. To correct for the effects of these parameter changes, we fit the part of the spectral trend produced by synaptic timescales using a FOOOF-like algorithm[14], except that the background trend was modelled using Eq. 1 (Fig. 7c; Fig. S4). As anticipated, detrending the spectra in this way corrected for the confounding effects of the parameter changes (Fig. 7d, e; Fig. S4).

In conclusion, our modelling results indicate that there are two distinct mechanisms of peak-trend interaction in EEG spectra. In one case, there are changes in the relative contribution of rhythmic and arrhythmic neural activity to the EEG signal. In this case, the peaks and spectral trend change relatively independently from one another, and thus detrending is unnecessary for quantifying spectral peak amplitudes (see Discussion). In the other case, changes in biophysical parameters alter the mechanism of EEG generation itself. In this case, EEG differences are unrelated to neural dynamics; these changes can confound EEG signals from all neural sources, and thus even spectral peak amplitudes can be potentially corrupted.

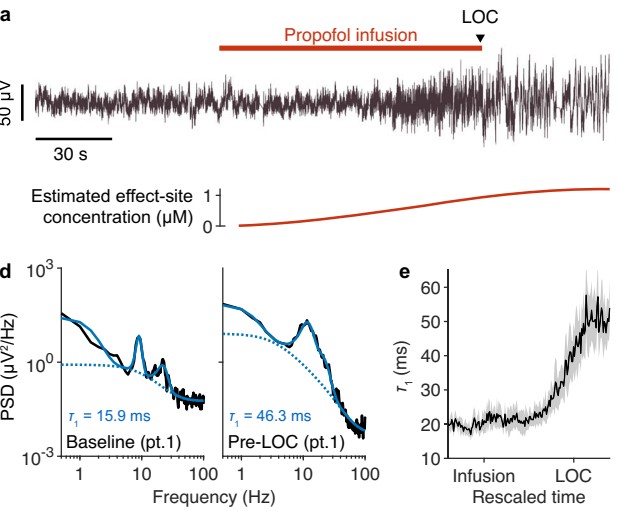

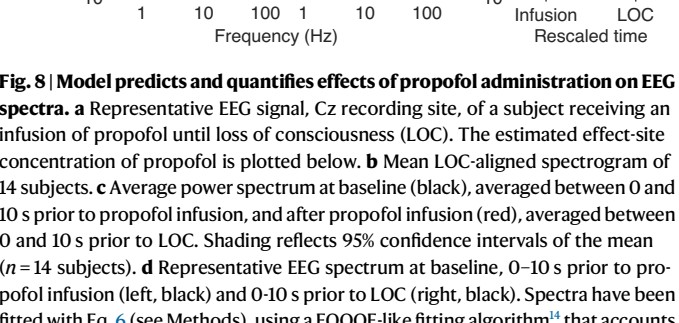

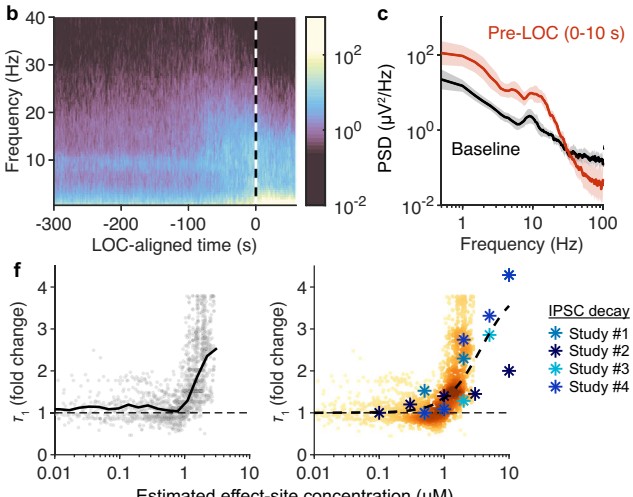

**Fig. 8 | Model predicts and quantifies effects of propofol administration on EEG spectra. a** Representative EEG signal, Cz recording site, of a subject receiving an infusion of propofol until loss of consciousness (LOC). The estimated effect-site concentration of propofol is plotted below. **b** Mean LOC-aligned spectrogram of 14 subjects. **c** Average power spectrum at baseline (black), averaged between 0 and 10 s prior to propofol infusion, and after propofol infusion (red), averaged between 0 and 10 s prior to LOC. Shading reflects 95% confidence intervals of the mean ($n = 14$ subjects). **d** Representative EEG spectrum at baseline, 0-10 s prior to pro-pofol infusion (left, black) and 0-10 s prior to LOC (right, black). Spectra have been fitted with Eq. 6 (see Methods), using a FOOOF-like fitting algorithm[14] that accounts for several Gaussian peaks in the spectra (solid blue). Fitted trend is shown with dashed line. **e** The parameter $\tau_1$, estimated from fits to spectra computed in 2-s windows, plotted with respect to rescaled time. Shading reflects 95% confidence intervals of the mean ($n = 14$ subjects). **f** Left: fold change in the estimated value of $\tau_1$ plotted against the estimated effect-site concentration of propofol. Black line marks the mean for each concentration of propofol. Right: The estimated dose-response plot from the left panel (darker colours reflect higher density of points) is superimposed with in vitro data taken from four studies: Study #1[27], Study #2[28], Study #3[29], and Study #4[29]. The data values taken from these studies are presented in Table S2. The dash black line is a fitted Hill function to the data from the four studies: $EC_{50} = 3.7\,\mu M$ and Hill coefficient ($n$) =1.6.

## Model predicts and quantifies effects of propofol on EEG spectra

The above results suggest that spectral detrending may be important in pharmacological experiments since many drugs target ion channels and alter postsynaptic responses. To test this model prediction, we investigated the EEG signatures of the general anesthetic propofol, a GABA$_A$ receptor modulator that lengthens the decay time of synaptic inhibition[27–30]. We recorded the EEG of 14 subjects during a fixed-rate infusion of propofol lasting until loss of consciousness (LOC) (Fig. 8a). The moment of LOC was identified by the dropping of a held object, which has been shown to provide an accurate, binary measure of LOC with precise timing[35,36]. The median LOC-aligned spectrogram was calculated from the Cz channel, which revealed an increase in low frequency power and a decrease in high frequency power starting before LOC (Fig. 8b). Comparing the average power spectrum at baseline (0–10 s prior to propofol infusion) to that following propofol infusion (0–10 s prior to LOC) revealed an increase in low frequency power and a decrease in high frequency power, thus giving the appearance of a rotation of the power spectrum (Fig. 8c).

Our modelling results suggest that propofol inflates low frequency power by increasing the slow timescale, $\tau_1$, of the spectral trend (Figs. 1d, e; 7). To test this, we estimated $\tau_1$ from the EEG data by fitting a modified Eq. 1 to the EEG spectra (Eq. 6; see Methods). This modified function fit the spectral trend well except at frequencies less than ~3 Hz (Fig. 8d; Fig. S5a–c), where our modelling results suggest the spectral trend is coloured primarily by neural dynamics, such as delta oscillations and/or subcritical network activity, but not synaptic kinetics (Figs. 4 and 5). Corroborating this interpretation, the low frequency (<3 Hz) part of the trend waxed and waned from second to second, occasionally disappearing entirely, whereas the fitted synaptic timescale remained stable between time windows (Fig. S5c). The extracted timescale, $\tau_1$, was remarkably consistent prior to the infusion of propofol, with an average value of $16.7 \pm 1.4$ ms (mean ± SE, $n = 14$) (Fig. 8e). Following the infusion of propofol, $\tau_1$ began increasing, reaching a value of $43.2 \pm 4.6$ ms ($p \approx 10^{-4}$; paired two-tailed $t$-test)

0–10 s prior to LOC (Fig. 8e). Similar changes were observed at other electrode sites (Fig. S6). By plotting the estimated value of $\tau_1$ at each time point against the estimated effect-site concentration of propofol, quantified using the Marsh model[37], we constructed a dose-response curve of fold change in $\tau_1$ against estimated propofol concentration (Fig. 8f). The inferred dose-response curve for $\tau_1$ quantitatively matched in vitro measurements of inhibitory postsynaptic current kinetics in the presence of bath applied propofol[27–29] (Fig. 8f). Thus, the changes in $\tau_1$ were consistent with expectations based on propofol's known pharmacology. These observations support the model's prediction that GABAR kinetics significantly shape EEG spectra and that broadband EEG changes do not necessarily reflect differences in brain dynamics.

## Correcting for synaptic timescales reveals a unique signature of LOC

Our modelling results predicted that propofol's effect on synaptic timescales will produce errors in conventional quantifications of brain rhythms. Past studies have suggested that LOC from propofol is associated with changes in the delta (0.5–3 Hz), alpha (8–15 Hz), and beta (15–30 Hz) frequency ranges[38]. To investigate the consequences of spectral detrending on EEG signatures of LOC, we compared raw bandpower to detrended bandpower in the delta, alpha, and beta frequency ranges. To compare EEG dynamics across individuals, data were aligned simultaneously to the moment of propofol infusion and LOC; this was done by rescaling time in each experiment by the latency to LOC (median: 135 s; range: 95–285 s). Similar results were obtained when time was not rescaled (Fig. S5d). Consistent with previous studies[38], raw baseline-normalized power increased in the delta, alpha, and beta band following propofol infusion (Fig. 9a–c). Whereas alpha power increased quickly and plateaued prior to LOC, delta power rose slowly and continued increasing until after LOC (Fig. 9d). Notably, beta power increased concomitantly with alpha power, but then began decreasing prior to LOC, eventually

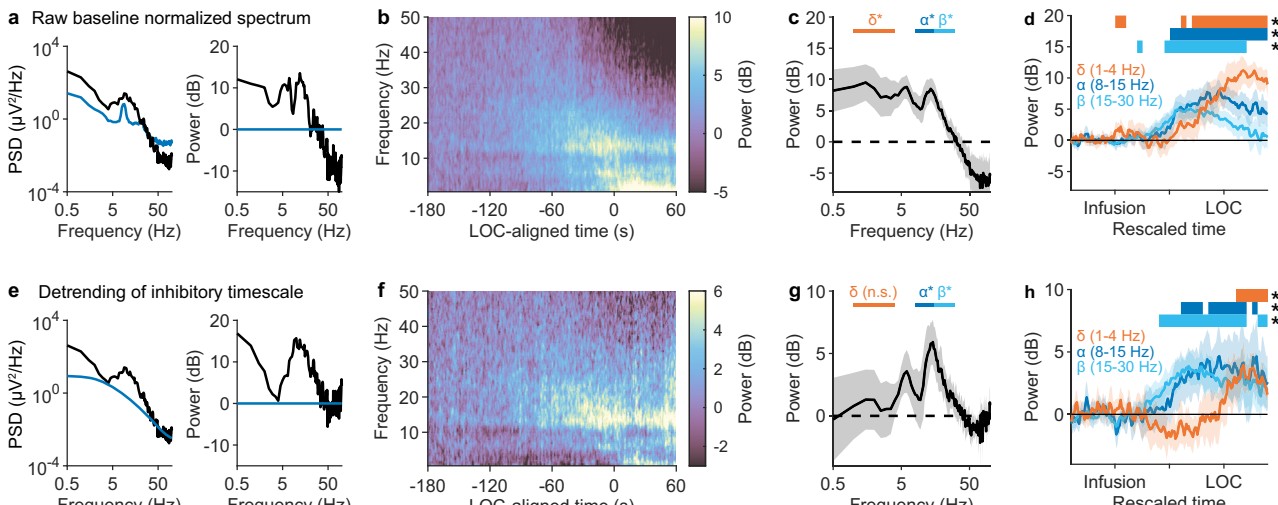

**Fig. 9 | Correcting for synaptic timescales reveals a unique signature of losing consciousness. a** Left: representative EEG power spectrum following LOC (black) of a single subject, superimposed on the power spectrum at baseline (blue). Right: power spectrum following LOC (black) normalized to baseline. **b** Mean baseline normalized spectrogram ($n = 14$ subjects). **c** Average baseline-normalized power 0–10 s prior to LOC (shading: 95% confidence interval of mean). Power was significantly elevated within the delta ($p = 0.001$, right-tailed sign test; $n = 14$ subjects), alpha ($p \approx 10^{-4}$), and beta ($p = 0.007$) frequency bands. **d** Changes in alpha, beta, and delta power, aligned to both moment of propofol infusion and LOC, averaged across subjects (shading: 95% confidence interval of mean). Bars above graph indicate 0.05-long segments of rescaled time where there is a statistically significant increase ($p < 0.05$, right-tailed sign test) in the corresponding frequency band power. Fig. S5d shows results without rescaled time. **e** Left: EEG spectrum following LOC, same as **a**, superimposed with fitted inhibitory timescale (Eq. 6). Fitted Gaussian peaks not shown. Right: detrended power, defined as power relative to fitted inhibitory timescale, in decibels. **f** Mean detrended spectrogram, normalized to baseline ($n = 14$ subjects). **g** Detrended power 0–10 s prior to LOC, normalized to baseline (shading: 95% confidence interval of mean). Power was significantly elevated within the alpha ($p = 0.006$) and beta, ($p = 0.001$), but not delta ($p = 0.40$) frequency bands. Significance testing same as (**c**). **h** Same as in (**d**), but for detrended power, normalized to baseline.

producing a beta power not statistically different from baseline post-LOC (Fig. 9d).

To correct for the confounding effects of propofol, we divided EEG power at each timepoint by the estimated synaptic timescales fitted from the previous section (Fig. 9e). We then normalized detrended power by the detrended power at baseline (Fig. 9f), such that changes in bandpower reflected the spectral changes unexplained by the increase in $\tau_1$. This procedure entirely removed the apparent rotation in the EEG spectra following propofol infusion (Fig. 9g). As a result of this detrending, power in the alpha, beta, and delta bands clearly increased at distinct times (Fig. 9h). Alpha and beta power appeared following propofol infusion and plateaued well before LOC. In contrast, detrended delta power did not increase until the moment of LOC, at which point it increased sharply (Fig. 9h; Fig. S5d). In summary, whereas the raw EEG signal failed to exhibit a strongly time-locked signal at the moment of LOC, removing the anticipated confounds of propofol revealed a sharp increase in delta power within seconds of LOC. This analysis combined with our modelling results suggest that pharmacological changes to synaptic kinetics may mask the true dynamics of neural oscillations when spectral detrending is not performed. In addition, these results point to distinct roles for alpha and delta power in the function of propofol as a general anesthetic.

## Discussion

In this study, we explored the neural basis of the EEG spectral trend and examined its implications for EEG interpretation and analysis. Several important conclusions came out of this investigation. First, this work provided biophysical evidence that arrhythmic neural activity is capable of generating detectable EEG signals. Second, our modelling consolidated the predictions of simpler computational models[16,20] and indicated that the EEG spectral trend is shaped by the interactions of many factors, including synaptic kinetics, excitatory-inhibitory ratio, and aperiodic network dynamics. Third, our analysis revealed that

spectral peak amplitudes are minimally affected by fluctuations in arrhythmic neural activity. On the other hand, we found that systemic changes in synaptic current properties do necessitate detrending to accurately interpret spectral changes as variations in neural activity. Our results suggested that this latter scenario is particularly important for EEG recordings used in tandem with pharmacological interventions.

Traditional spectral analysis assumes that EEG power within canonical frequency bands reflect various brain rhythms. Our modelling results seriously challenge this assumption. Specifically, if a spectral peak is not evident within a given frequency band, our work justifies a physiologically plausible and biophysically realistic alternative hypothesis, namely, that the bandpower reflects broadband neural activity. Our results thus validate the principle assumption of spectral detrending methods, such as the FOOOF algorithm[14], which is that peak detection is a necessary prerequisite to quantifying oscillatory power. While power lying outside spectral peaks could potentially reflect neural rhythms, there is no theoretical guarantee for this interpretation based on spectral analysis alone.

Importantly, our results do not necessarily support a purely data-driven approach to detrending EEG spectra. While changes in arrhythmic activity affect spectral peaks in an additive manner (Fig. 5), changes in synaptic currents multiplicatively scale spectral peaks (Fig. 7); these two mechanisms thus require two distinct methods of detrending, i.e., subtractive versus divisive detrending, respectively. Without prior knowledge, it is unclear which method is required. Moreover, given the amplitude of peaks above the background trend typical in EEG spectra, small additive changes in the trend would alter the peak amplitudes minimally compared to the errors introduced by incorrectly detrending (Fig. 5). Overall, we conclude that spectral detrending should not be performed unless there is clear physiological and biophysical justification and validation.

In the analysis presented here, there was prior knowledge of the well-documented action of propofol on GABAR kinetics[27–29], clearly

indicating the necessity for divisive detrending. To mitigate the chances of overfitting, we constrained parameter values to physiologically reasonable ranges, and validated that the fitted spectral changes and $\tau_1$ values quantitatively matched expectations. Even so, our decomposition of spectra into physiologically distinct components was likely imperfect, especially considering that changes to GABAR kinetics also probably affected aperiodic network dynamics[39] (see below). Although inhibitory synapses and network timescales are expected to affect spectra in different frequency ranges (Figs. 1, 4, 5, 7), these ranges overlap, making it challenging to definitively disambiguate these two mechanisms. Such caveats may be addressable with future refinements to EEG detrending algorithms, e.g., by including phase information or using generative models to improve parameter inference[40].

Past modelling has suggested that the EEG spectral exponent reflects the E:I ratio, which relies on the assumption that low frequency power is dominated by inhibitory currents[16]. Our results broadly validate this assumption, suggesting that the E:I ratio is an important driver of the spectral trend. However, our results also demonstrate that the spectral exponent is not a reliable measure of E:I ratio because of nonlinearities in membrane potential dynamics related to the reversal potentials of AMPARs and GABARs (Fig. 6), an aspect not considered in previous work[16]. In addition, we found that other biophysical parameters, such as synaptic timescales and leak currents contribute to shaping the EEG spectral trend. These findings are consistent with past studies that have associated neuronal timescales with the expression levels of various ion channels and receptor subunits[41]. It seems likely that changes in the spectral trend can reflect a large number of physiological mechanisms that converge to govern synaptic kinetics and the effective E:I ratio. Other biophysical mechanisms not explored here, such as active membrane currents[42] and dendritic calcium spikes[43], could also potentially contribute to shaping EEG spectra by changing postsynaptic response kinetics and amplitudes. Broadly, we conclude that the spectral exponent reflects physiological factors other than neural oscillations and is therefore a complementary biomarker to brain rhythm quantification.

While several studies have suggested that avalanche criticality may be responsible for broadband $1/f^\beta$ scaling in EEG spectra[8,18,21,44], our results corroborate other models of recurrent neural networks[20] and indicate that avalanche criticality would likely contribute to only slower frequency components of EEG (Fig. 4, in the limit as $m \rightarrow 1$). In a general sense, our model suggests an underlying mechanism for the observed 1/f trend comparable to that proposed for how purely oscillatory networks might produce such a trend. In the oscillatory case, it has been suggested that large networks oscillate slower than smaller networks, thus leading to a 1/f trend[22,23]. Our results suggest that propagating spikes that exhibit longer correlation timescales can produce more coherent dipoles, thus contributing more to the EEG signal. This in turn would cause EEG spectra to be dominated by lower frequency aperiodic signals. While still technically challenging, our model could be tested by extending recent work characterizing the functional clustering of synapses within individual dendritic branches[45–49], and by measuring the geometric relations between correlated synapses across neighbouring neurons.

Delta rhythms are a ubiquitous feature of general anesthesia, being readily induced in humans by propofol, sevoflurane, thiopental, and xenon[38,50–55], potentially indicating a universal signature of unconsciousness bridging both general anesthesia and sleep[56–58]. To the best of our knowledge, such an abrupt change in delta power as reported here has not been previously reported in EEG studies, even when the moment of LOC was resolved with high temporal precision[38]. Although practically detrending power relies on simplified models of EEG spectra, our analysis clearly shows that the slow changes in delta power prior to LOC can be explained by propofol's action on GABAR kinetics, and thus demonstrates that delta power originating from

changes in neural dynamics appears after the propofol-induced increase in alpha and beta power (Fig. 9). According to our simulations, it is possible that the observed increase in low frequency power is due to the brain moving closer to avalanche criticality (Figs. 4 and 5). Opposing this interpretation are models of excitatory/inhibitory networks, which suggest that increasing inhibition promotes the asynchronous state[39]. Furthermore, detailed analyses of EEG signals have suggested that altered states of consciousness are related to a departure away from, and not towards, critical dynamics[59,60]. Based on these studies, we should rather expect changes in network criticality that decrease delta power. It seems likely, therefore, that the increase in delta power at the moment of LOC is related to changes in delta rhythms. If so, our observations lend evidence to the view that delta rhythms are fundamental to losing consciousness. A target effect-site concentration protocol could be used to investigate lower doses of propofol in more detail, which our observations suggest may induce alpha rhythms but not delta rhythms. If so, these experiments could be used to dissect the behavioural correlates of these two rhythms in more detail.

In summary, we conclude that aperiodic neural activity can contribute to EEG signals, and that the spectral trend is further shaped by many physiological mechanisms, such as excitatory/inhibitory balance and synaptic timescales. We conclude that the spectral exponent is not merely a conflated measure of brain rhythms and thus provides a complementary biomarker of brain state. However, we also conclude that the spectral exponent does not have a singular physiological interpretation. Finally, we conclude that EEG spectra do need to be detrended when quantifying brain rhythms, but only if postsynaptic current properties are systematically altered. Otherwise, detrending likely introduces significant errors to brain rhythm quantification and should therefore be avoided.

## Methods
### Definitions and theoretical framework
The EEG signal can be described as the linear superposition of electric fields generated by all neurons in the brain[3,61]. We refer to the individual contribution of a single neuron to the ensemble EEG signal as a single-neuron EEG. Otherwise stated, the ensemble EEG signal is the linear summation of $N$ single-neuron EEGs. This means that the power spectral density of an EEG signal can be expressed as

$$S_N(f) = \sum_i S_i(f) + 2\sum_{i<j} \gamma_{ij}(f)\sqrt{S_i S_j}, \qquad (2)$$

Here, $S_i$ is the power spectrum of the single-neuron EEG generated by a given neuron $i$, and $\gamma_{ij}$ is the coherence between the single-neuron EEG of neuron $i$ and neuron $j$. Spectral peaks are thought to appear due to coherence at a given frequency[3]. Because the coherence function, $\gamma_{ij}$, interacts multiplicatively with single neuron EEG spectra, the spectral features of the ensemble EEG will be governed in part by the spectral features imparted by neurons at the single cell level. Moreover, this influence will be independent of the nature of brain dynamics, except insofar as these dynamics alter the single-neuron EEG spectrum. Broadly, this paper concerns itself with investigating how broadband single-neuron EEG signals influence the spectral features of the ensemble EEG.

### Postsynaptic neuron simulations
Neuron morphologies as well as their relative abundance were the same as those used by Hagen et al.[25] (Table S1). The same biophysical parameters were used for all neuron subtypes. For simulations, morphologies were segmented such that each compartment was less than 10 μm. Each postsynaptic neuron was modelled with an axial resistance $R_a = 100\ \Omega$ cm and a membrane capacitance of 1 μF cm$^{-2}$. All compartments were passive. The maximal leak conductance ($g_L$) was

investigated in a range from 0.01 to 5 mS cm$^{-2}$ and the leak reversal potential ($E_L$) was investigated in a range from $-75$ to $-45$ mV (Fig. 1g). The number of synapses for each cell was equal to the total dendritic length times a density of 1 synapse per μm for excitatory synapses and 0.15 synapses per μm for inhibitory synapses[62–64]. Synapses were distributed among all compartments proportionally to the compartments' surface area. Post-synaptic currents were modelled as the difference of exponentials. Excitatory synapses had a reversal potential of 0 mV, a rise time of 0.3 ms, a decay time constant ($\tau_E$) between 1 and 3.5 ms, and a peak conductance ($g_E$) between 0.2 and 2 nS (Fig. 1g). Inhibitory synapses had a reversal potential of $-80$ mV, a rise time of 2 ms, a decay time constant ($\tau_I$) between 5 and 20 ms, and a peak conductance ($g_I$) between 0.2 and 2 nS (Fig. 1g). For illustrating specific example spectra, e.g., those shown in Figs. 1d, 2f, 4f, g, 5 and 7, the following parameter set was used: $g_L = 1$ mS cm$^{-2}$, $E_L = -58$ mV, $\tau_E = 1.8$ ms, $g_E = 0.7$ nS, $\tau_I = 10$ ms, and $g_I = 0.7$ nS. All simulations of neurons were performed in Python 3.8.10 using the package LFPy 2.2.4[65], running the NEURON simulation environment under the hood[66].

## Ensemble EEG amplitude estimation

Because dipoles sum linearly, the EEG signal generated by N neurons can be decomposed as a superposition of N single-neuron EEG signals. It follows that the ensemble EEG will have an average power of $\sigma_N^2 = \sum_{i=1}^N \sigma_i^2 + 2\sum_{i<j} \rho_{ij}\sigma_i\sigma_j$, where the single-neuron EEG signals produced by neurons $i$ and $j$ have average powers of $\sigma_i^2$ and $\sigma_j^2$, respectively, and a pairwise correlation of $\rho_{ij}$. Dipole correlations should arise if two conditions are satisfied: (1) synaptic inputs are correlated, which we assumed to decrease with distance; and (2) dipole orientations are aligned, which we assumed depends on both condition 1 and on the angle between the apical-basal axes of the neurons. We therefore modelled the dipole correlation of two neurons as $\rho_{ij} = \exp(-d_{ij}^2/\sigma^2)\cos\theta_{ij}$, where $d_{ij}$ is the distance between the two neurons, $\sigma$ is some characteristic spatial scale of correlation, and $\theta_{ij}$ is the angle between the apical-basal axes of the respective neurons. We estimated these parameters using the ICBM152 v6 anatomical brain template[24,67,68], assuming a uniform density of $\mu = 100{,}000$ neurons per mm$^2$ of cortical surface area[69].

The brain template is a triangular mesh with faces of areas $A_j$ and normal vectors $\vec{N}_j$. Given a random point, $x_i$, on the mesh, we used the vertex points of the mesh to analytically calculate the "signed number" of neurons within a given radius $r$, given by

$$\nu_i(r) = \sum_j \mu\left(A_j f_j(r, x_i)\right)\vec{N}_j \cdot \vec{N}_i, \tag{3}$$

where $f_j(r, x_i)$ is the fraction of triangular mesh face $j$ that intersects a ball of radius $r$ centred at point $x_i$. We repeated this $k = 2000$ times with different starting points, $x_i$, to estimate an average $\bar{\nu}(r) = \frac{1}{k}\sum_k \nu_i(r)$. This allowed us to estimate the average pairwise correlation across the entire cortex with the following formula

$$\bar{\rho} = \frac{1}{N-1}\int_{r>0} \exp(-r^2/\sigma^2)\,d\bar{\nu}(r), \tag{4}$$

It follows that the expected signal power of the ensemble EEG signal is $\sigma_N^2 = N\sigma_0^2 + N(N-1)\bar{\rho}\sigma_0^2$, where $\sigma_0^2$ is the expected power of a single-neuron EEG (Fig. 1).

Experimental quantification of $\sigma^2$ would require measuring and comparing the isolated dipoles of individual neurons, a procedure that is not possible. However, correlations in subthreshold membrane potentials have been investigated in cats, both in the presence and absence of oscillations[70]. These experiments revealed that in the absence of oscillations, subthreshold fluctuations were correlated between neocortical neurons up to ~5 mm apart, but were uncorrelated between neurons ~13 mm apart[70]. Because dipoles are predominantly generated by subthreshold currents[4,71], these data suggest a possible lower and upper bound for $\sigma^2$.

## Subcritical network model

Presynaptic neurons were connected to $d_{out} = 10$ other neurons in the presynaptic network. The probability of two neurons being connected was determined by their distance, using an exponentially decaying coupling kernel. This rule forced network connectivity to be local in nature. Each neuron followed a Poisson point process with a rate $\lambda_E(1 - m)$, where $\lambda_E$ was sampled from the distribution shown in Fig. 1g. When a neuron spiked, each of its neighbouring neurons had a probability of $m/d_{out}$ of firing an action potential within the following 4 ms[32]. The parameter $m$ thus tuned the amount of spike propagation in the network, changing the network behaviour from completely asynchronous when $m = 0$, to near avalanche criticality as $m$ approached 1. The above formalism was used to simulate excitatory presynaptic neurons. We also added inhibitory neurons totalling 15% of the entire network. Spike trains for inhibitory neurons were sampled from the spike trains of nearby excitatory neurons and supplemented with independent Poisson processes. The procedure was such that inhibitory neuron firing was driven by recurrent connections to the same proportion as excitatory neurons and followed a predetermined firing rate ($\lambda_I$). In this setup, the influence of inhibitory neurons on the network was modelled implicitly in the branching number[39], but were not explicitly represented in the network connections. This simplification allowed the effective branching number of the network to be entirely governed by a single parameter $m$.

## Embedding synapses onto postsynaptic dendrites

Dipole synchrony has been previously modelled either by separating inhibitory and excitatory input into somatic and apical compartments, respectively[3], or by having counterphase input into the basal and apical compartments[72,73]; both of these models can be thought of as optimal mappings of two anticorrelated populations of synapses. Inspired by these models, we developed a procedure to generate dipole synchrony given any presynaptic topology. First, the pairwise correlation between each pair of presynaptic neurons was determined by the spike time tiling coefficient (STTC)[74], a measure of spike train correlation which accounts for the likelihood of spikes overlapping by chance. The pairwise correlations among all presynaptic neurons were computed based on 40 s long simulations of the presynaptic networks. The Uniform Manifold Approximation and Projection (UMAP) algorithm, a dimensionality reduction technique[31], was then used to optimally project the presynaptic network onto a sphere such that the angle between presynaptic neurons with high STTCs was minimized. After this embedding step, the dendrites of the postsynaptic neurons were orthogonally projected onto the sphere. Finally, the following procedure was run until all connections were formed between pre- and post-synaptic neurons: (1) a postsynaptic dendrite segment was chosen randomly (with replacement) with a probability proportional to its surface area; (2) the presynaptic neuron closest on the spherical embedding was chosen and a connection formed.

To model suboptimal synapse placement, we randomly perturbed the spherical embedding before mapping the synapses. Each point in the embedding was perturbed by a distance $\pi \arccos(1 - 2\alpha(1 - X))$ along a randomly chosen bearing, where $\alpha \sim \text{Uniform}[0,1]$. Here, $X$ is what we refer to as the optimality index (Fig. 4e). By construction, when $X = 1$, the points on the sphere are not perturbed at all, while when $X = 0$, all the points on the sphere are perturbed by a distance sampled from the function $\pi \arccos(1 - 2\alpha)$. Consequently, for $X = 0$, the distribution of points on the sphere is uniformly random.

This procedure can be thought of as a model of dipole correlation that generalizes beyond dichotomous input. Alternatively, this procedure can be considered in terms of observations from recent

studies, which have reported that functionally related synaptic inputs cluster within individual dendritic branches, and that input from similar presynaptic populations target similar dendritic compartments in the postsynaptic population[45–49]. These experimental observations hint at a more continuous mapping of synaptic input along dendrites than a strictly binary apical-basal compartment paradigm, which is precisely what is achieved when the above mapping algorithm is applied to a continuous presynaptic network topology, such as the planar subcritical network used in Fig. 5.

### Mixed synaptic input

For modelling oscillatory input, we used a published formalism of rhythmogenesis, where counterphase sinusoidal inputs were applied on the apical and basal dendrites[72,73]. Specifically, every synapse on the neuron received an inhomogeneous Poisson point process as input, with a rate function $\lambda_x(1 + \alpha_R \sin(2\pi\omega t + k))$, where $\lambda_x$ depends on whether the synapse is excitatory $(\lambda_E)$ or inhibitory $(\lambda_I)$, $\alpha_R$ tunes the strength of the rhythm, and $k = \pi$ for apical dendritic synapses and $k = 0$ for basal dendritic synapses. For simplicity, to model other rhythms, we generalized this formalism by defining the rate function of each synapse as $\max(\lambda_x(1 + \alpha\widetilde{Y}(t)), 0)$, where $\widetilde{Y} = Y(t)$ if the synapse was an apical synapse and $\widetilde{Y} = 1 - Y(t)$ if the synapse was a basal synapse, for any time series $Y(t)$ with zero mean and unit variance. This methodology was simpler than modelling networks for each type of dynamic and embedding the synapses as described in the above section, but more importantly this methodology allowed us to make direct comparisons between the modelled EEG signals generated by different rate functions.

### Experimental design and procedure

Following MNH Ethics Board approval, we recruited 16 American Society of Anesthesiologists (ASA) class I or II patients (18–65 years old) presenting for lumbar disk surgery as subjects for the study. All subjects gave written informed consent to participate in the study. The standards of care of the Canadian Anesthesiologists' Society in regard to monitoring, equipment and care provider were rigorously applied. Gold cup electrodes (Fz, Cz, Pz, C3, C4, CP3, CP4, M2 as reference; FC1 as ground; impedance ≤ 5 kOhm) were glued to the scalp to obtain a continuous EEG recording, which was amplified with a 0.1–300 Hz band pass and digitized at 1024 Hz. For each participant, we obtained 2 min of recording during preoxygenation with eyes closed. The participant was then asked to hold an object (0.5 kg cylinder; 2.5 cm diameter and 15 cm long) in a vertical position with their dominant hand and to keep the eyes closed. Preoxygenation continued for another 2 min. Lidocaine 2% (40 mg) was given to attenuate the discomfort caused by the propofol injection. All medications were given intravenously via a catheter placed on the non-dominant arm. Propofol was given at the rate of 1 mg kg$^{-1}$ min$^{-1}$ and maintained until the cylinder fell from the participant's hand. Following LOC, gentle jaw lift was applied if needed to relieve airway obstruction. The ability of the participants to respond to loud verbal command was assessed 60 seconds after the fall of the object: all failed to response and remained immobile. The study was then terminated.

### Data analysis

Two participants were excluded because of failure to comply with the instructions during induction. One kept talking, the other kept moving their dominant arm. The final data set was therefore based on 14 subjects (10 males; 12 right-handed). Artifacts in the data were removed following visual inspections of the time series. Spectrograms were computed with the multitaper method, using three tapers over 2 s windows, with 1.9 s overlap. Group averages were either computed as the average power spectral density across subjects with time aligned to LOC, or with time rescaled so that both the infusion onset and LOC were aligned across individuals. To rescale time, LOC-aligned time for each subject was scale by the latency from infusion to LOC.

Consequently, a rescaled time value of −1 is equivalent to the moment of propofol infusion onset and a rescaled time value of 0 is equivalent to the moment of LOC.

### Estimated propofol concentration

For estimating the effect-site concentration of propofol for each subject, we used the Marsh model, a multicompartment pharmacokinetics model[37]. We used a plasma to effect-site equilibration rate constant $k_{eo} = 1.21$ min$^{-1}$ [75]. Faster and slower values for $k_{eo}$ would shift the estimated dose-response curve in Fig. 8f right and left, respectively, but would not be expected to qualitatively change our results.

### Detrending EEG spectra

To detrend EEG spectra, we used a modified version of the FOOOF algorithm[14], whereby a background trend is fitted in addition to several Gaussian functions to account for oscillatory peaks. Peaks at 60 Hz due to noise were removed from spectra using MATLAB's fillgaps function prior to fitting the spectral trend. The FOOOF algorithm provides two options for the background trend, $A/f^\beta$ or $A/(k + f^\beta)$. Here, we wanted a biophysically interpretable background function, and therefore started with the sum of two Lorentzian functions (Eq. 1). This function is an exact analytical solution to the computational model of Gao et al. [16] and has several benefits. Firstly, it provides exact fits to our simulations, which allowed us to investigate theoretical consequences of detrending EEG spectra. Secondly, the parameters are physiologically interpretable. $\tau_1$ and $\tau_2$ reflect the kinetics of GABARs and AMPARs (Fig. 1), while $A_1$ and $A_2$ reflect the amplitudes of inhibitory and excitatory contributions to the EEG signal. Thirdly, it is thought that the spectral exponent, $\beta$, reflects the relative contribution of excitation to inhibition[16], i.e., it depends on the ratio of $A_1$ to $A_2$ (Fig. 1). However, we found here that this is not always the case (Fig. 6). Therefore, Eq. 1 makes fewer assumptions about its parameters than the two options provided by the FOOOF package.

For analyzing experimental data, we modified Eq. 1 to provide better fits to the data and reduce the chances of overfitting. In contrast to our simulations, we found that high frequency power plateaued in our data around where we would expect the influence of excitatory synaptic time scales to be exerted (Fig. 1d, f). We therefore replaced the second term in the equation with a constant term,

$$A_1\tau_1 / \left(1 + \left(2\pi\tau_1 f\right)^2\right) + \lambda. \tag{5}$$

This constant term, $\lambda$, captures the fast excitatory time scales as well as any high frequency contributions from action potentials[4], muscle activity[76], and amplifier noise[7], which were not present in our simulations. Importantly, because this equation has physiologically interpretable parameters, with $\tau_1$ reflecting inhibitory synaptic time scales (Fig. 1d, e), we could constrain the range of $\tau_1$ to avoid overfitting. Specifically, we constrained $\tau_1$ to be greater than 10 ms and less than 75 ms when fitting propofol data, as we expected propofol to increase the physiological range of GABAR kinetics (Table S2).

This modified equation fit the EEG spectra at baseline conditions well, but following propofol infusion, the equation did not decay fast enough to capture the EEG spectra (Fig. S7). This was seemingly because the original equations oversimplified the kinetics of inhibitory synapses. Notably, Eq. 1 is an analytical solution for exponentially decaying synaptic responses, whereas real synaptic responses are characterized by a rise time and decay time: $\exp(-t/\tau_1) - \exp(-t/\tau_r)$. It follows that a more accurate synaptic response function for the power spectrum is given by

$$\frac{A_1(\tau_r - \tau_1)^2}{\left(1 + \left(2\pi\tau_r f\right)^2\right)\left(1 + \left(2\pi\tau_1 f\right)^2\right)} + \lambda, \tag{6}$$

where the first term is the analytical solution to the power spectrum of the difference of exponentials. $\tau_r$ is the risetime of inhibitory synaptic currents, which we fixed at $\tau_r = 4$ ms to keep the number of fitting parameters low[77]. Notably, if we fixed both $\tau_r = 4$ ms and $\tau_1 = 20$ ms, physiologically plausible values[77], all our baseline data could be captured by simply changing $A_1$ and $\lambda$ (Fig. S5a). Moreover, fitting Eq. 6 provided consistent and physiologically plausible estimates for $\tau_1$ both at baseline and following propofol infusion (Fig. 8e, f). Thus, Eq. 6 is biophysically motivated, physiologically interpretable, has only three parameters to fit, and captured our data well both at baseline and following the infusion of propofol (Fig. 8d; Fig. S5a–c).

## Reporting summary

Further information on research design is available in the Nature Portfolio Reporting Summary linked to this article.

## Data availability

Computed EEG spectrograms for all subjects have been uploaded to Figshare[78] (https://doi.org/10.6084/m9.figshare.24777990), along with all the simulation results required to reproduce our figures. Source data are provided with this paper.

## Code availability

Code used to run simulations, analyze data, and generate manuscript figures[79] has been deposited in Zenodo (https://doi.org/10.5281/zenodo.10359818) and is also available on GitHub (github.com/niklasbrake/EEG_modelling).

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

## Acknowledgements

This work was supported by the Natural Sciences and Engineering Research Council of Canada (NSERC) discovery grant (RGPIN-2019-04520) to A.K. N.B. was supported by the NSERC-CREATE in Complex Dynamics Graduate Scholarship and the Fonds de recherche du Québec

– Nature et technologies (FRQNT) doctoral training scholarship. The funders had no role in study design, data collection and analysis, decision to publish, or preparation of the manuscript.

## Author contributions

Conceptualization: N.B., G.P. and A.K. Data curation: N.B. and G.P. Formal analysis: N.B. Funding acquisition: A.K. and G.P. Investigation: N.B., F.D., A.R., F.A., S.S. and G.P. Methodology: N.B. Project administration: A.K. and G.P. Software: N.B. Supervision: A.K. and G.P. Visualization: N.B. Writing—original draft: N.B. Writing—review and editing: N.B., A.K. and G.P.

## Competing interests

The authors declare no competing interests.
