## [Peer Review File · Nature Communications]

A neurophysiological basis for aperiodic EEG and the background spectral trendEditorial Note: This manuscript has been previously reviewed at another journal that is not operating a transparent peer review scheme. This document only contains reviewer comments and rebuttal letters for versions considered at *Nature Communications*.

REVIEWERS' COMMENTS

Reviewer #2 (Remarks to the Author):

The Authors substantially revised the manuscript, which now reads much better and it is also much clearer conceptually speaking. My main concerns were sufficiently addressed and I only have some minor comments remaining.

1. Line 191, 'the spectral trend interacted approximately linearly with the oscillatory peak amplitude'. In the end I could understand what the Authors mean by 'interacted linearly', but it was not clear at first; e.g. it could also imply some additive relationship. I suggest rephrasing this statement (same in line 246).

2. Fig.5f 'unitary spectrum from _a_ was divided by the solid grey line. Please be more precise here, as it is not entirely clear what was divided by what. I don't see a unitary spectrum in panel a. Similarly, Fig.7d 'dividing the spectrum by the fitted trend' could use some clarification.

These minor points aside, the manuscript clearly demonstrates an interesting model that have implications for interpreting EEG spectra, therefore potentially interesting to many of us.

I believe that openness and transparency increase the fairness of peer review. Therefore, I decided to sign my reviews.

Balazs Hangya

Reviewer #3 (Remarks to the Author):

Many thanks to the authors for constructively engaging with my comments on the original manuscript. I think the revised version has a much more coherent message with a clear narrative throughout which builds one section at a time. More importantly, after reframing of the question and results, its main claims are well-supported by evidence, and the additional analyses and figures contain some very interesting new results (such as Figure 4f).

Overall, my comments from the original manuscript still hold: I enjoyed reading the paper a lot, the revised manuscript even more so. I believe it will be a key contribution to the literature on understanding the aperiodic component of the EEG, as well as biophysical modeling of EEG in general, combining many different “classes” of models in a thoughtful way. Also, I have to comment once again on the fantastic quality of the figures.

I have just a few minor points that I hope for the authors to clarify / discuss:

- One of the few points in the paper I found confusing is the conclusion that “spectral detrending”, i.e., removing the $1/f$ background before measuring narrowband amplitude, simultaneously 1) should be avoided because it produces erroneous or “misleading results” (e.g., Fig 5 caption), 2) unnecessary because “spectral peaks are minimally affected” (L314), and 3) is necessary because “systematic changes in synaptic changes alters EEG generation” (L316). I’m not saying I disagree with the results, it was just hard to follow what exactly the recommendation was and why.

- If I understand correctly, there are essentially 3 ways the spectrum can change: neural dynamics, which the authors separate into oscillatory and aperiodic/asynchronous and model as separate inputs into the biophysically accurate neuron, and synaptic currents, which is a biophysical property of the neuron itself. Figures 5 and 7 combine to show that oscillatory and aperiodic input do not adversely affect each other in the spectrum, and that detrending the aperiodic background in the typical “FOOOF-like” fashion can underestimate oscillatory drive (Fig 5f), whereas synaptic amplitude / time constant changes can alter the spectrum in a way that requires detrending. Obviously, in real data, one cannot be sure a priori whether a change in the aperiodic part of the spectrum is due to aperiodic change or synaptic change, or both (see point below). So how does one take these insights from simulations into analyzing real experimental data? I appreciate that these findings demonstrate that the underlying issue is complicated and situation-dependent, and therefore difficult to communicate clearly. So this is not a criticism, just feedback that I was a bit confused as a reader.

- For the analysis in Figure 5f-h, is the error in detrended spectral peak power measurement introduced because log power (i.e., in units of dB) is considered? If one first fit the aperiodic component using Equation 2 (or use any such FOOOF-like procedure), and then subtract the aperiodic fit in linear power instead (i.e., in units of V^2/Hz), would this result in a consistent measurement between 5f and 5h? In any case, I think 5d&g raises an interesting question of whether the measured oscillatory power decrease after detrending is meaningful, and is worth emphasizing more.

- Throughout the paper, if I understand correctly, neural dynamics is simply considered as input into the biophysical neuron to compute the unitary spectra, and is independent from the synaptic factors. As already discussed in the previous round of review, while this is fine for forward simulating the contribution of each on the EEG spectrum, especially to exert independent experimental control, this is rarely decoupled in real neural circuits. If we assume that the neurons that produce the EEG is a local filter shaped by its neuronal and synaptic biophysical properties, and that it does not participate in a circuit with its own neuronal dynamics, as the authors do here, then the forward simulation and reverse

inference are consistent with each other. However, this rarely the case, changing synaptic time constants and amplitudes will change the local recurrent neural dynamics, this is well known. In fact, the mechanism through which propofol acts is likely that the local and global neural dynamics are altered due to a change in the GABA synaptic properties.

- I would emphasize that this is a point that should be discussed as a limitation of both the model, and the interpretation of the EEG data (and any real data), in the discussion section. I don't think more extensive modeling is required from the authors, but the fact that these different factors mix together in their causal impact on the final measured EEG is exactly the reason reverse inference is so difficult should be communicated, and extending the modeling study to consider this would be an exciting future direction.

- as per the reviewer response, I think showing the tau change for all the channels would be great as a supplemental figure (Response 4 under reviewer 3 in the response letter)

Richard Gao, PhD

University of Tuebingen

REVIEWERS' COMMENTS

Reviewer #2 (Remarks to the Author):

1. Line 191, 'the spectral trend interacted approximately linearly with the oscillatory peak amplitude'. In the end I could understand what the Authors mean by 'interacted linearly', but it was not clear at first; e.g. it could also imply some additive relationship. I suggest rephrasing this statement (same in line 246).

Response 1: As suggested by the reviewer, we have rephrased these statements to further clarify our message. The first statement now reads, "However, changes to the spectral trend did not multiplicatively scale the oscillatory peak amplitude. As a result, quantifying the amplitude of peaks relative to the spectral trend produced incorrect interpretations" (line 199). The second statement now reads, "In this case, the peaks and spectral trend change relatively independently from one another, and thus detrending is unnecessary for quantifying spectral peak amplitudes" (line 259).

Additionally, we have elaborated on the linearity/additive nature of the interaction in more detail in the Discussion (lines 347-354) while addressing Reviewer #3's remarks below.

2. Fig.5f 'unitary spectrum from _a_ was divided by the solid grey line. Please be more precise here, as it is not entirely clear what was divided by what. I don't see a unitary spectrum in panel a. Similarly, Fig.7d 'dividing the spectrum by the fitted trend' could use some clarification.

Response 2: We thank the reviewer for bringing this up and we do apologize for the confusion that arose from a typo. The caption now correctly reads, "The unitary spectra in c were divided by the solid grey line." We have also clarified the caption of Fig. 7d.

Reviewer #3 (Remarks to the Author):

1. One of the few points in the paper I found confusing is the conclusion that "spectral detrending", i.e., removing the $1/f$ background before measuring narrowband amplitude, simultaneously 1) should be avoided because it produces erroneous or "misleading results" (e.g., Fig 5 caption), 2) unnecessary because "spectral peaks are minimally affected" (L314), and 3) is necessary because "systematic changes in synaptic changes alters EEG generation" (L316). I'm not saying I disagree with the results, it was just hard to follow what exactly the recommendation was and why.

If I understand correctly, there are essentially 3 ways the spectrum can change: neural dynamics, which the authors separate into oscillatory and aperiodic/asynchronous and model as separate inputs into the biophysically accurate neuron, and synaptic currents, which is a biophysical property of the neuron itself. Figures 5 and 7 combine to show that oscillatory and aperiodic input do not adversely affect each other in the spectrum, and that detrending the aperiodic background in the typical "FOOOF-like" fashion can underestimate oscillatory drive (Fig 5f), whereas synaptic amplitude / time constant changes can alter the spectrum in a way that requires detrending. Obviously, in real data, one cannot be sure a priori whether a change in the aperiodic part of the spectrum is due to aperiodic change or synaptic change, or both (see point below). So how does one take these insights from simulations into analyzing real experimental data? I appreciate that these findings demonstrate that the underlying issue is complicated and situation-dependent, and therefore difficult to communicate clearly. So this is not a criticism, just feedback that I was a bit confused as a reader.

For the analysis in Figure 5f-h, is the error in detrended spectral peak power measurement introduced because log power (i.e., in units of dB) is considered? If one first fit the aperiodic component using Equation 2 (or use any such FOOOF-like procedure), and then subtract the aperiodic fit in linear power instead (i.e., in units of V^2/Hz), would this result in a consistent measurement between 5f and 5h? In any case, I think 5d&g raises an interesting question of whether the measured oscillatory power decrease after detrending is meaningful, and is worth emphasizing more.

Response 1: As requested by the reviewer, we have now clarified these points in our Discussion (lines 347-354). Broadly, we have outlined two mechanisms that affect spectral trend: one that interacts with peaks linearly (Fig. 5) and another that interacts multiplicatively (Fig. 7).

Indeed, the reviewer is correct that subtracting the aperiodic fit instead of detrending divisively would produce consistent results between Figs. 5f and 5h. However, this analysis would then be incorrect when applied to the simulations of Fig. 7. When analyzing real data, one does not know which effect is taking place without prior knowledge, hence our recommendation that detrending not be performed without biophysical justification (as we have now highlighted more directly in the Discussion). To clarify our recommendations, we now include a summary of our conclusions at the end of our Discussion that relate to practical EEG interpretation (413-420).

2. Throughout the paper, if I understand correctly, neural dynamics is simply considered as input into the biophysical neuron to compute the unitary spectra, and is independent from the synaptic factors. As already discussed in the previous round of review, while this is fine for forward simulating the contribution of each on the EEG spectrum, especially to exert independent experimental control, this is rarely decoupled in real neural circuits. If we assume that the neurons that produce the EEG is a local filter shaped by its neuronal and synaptic biophysical properties, and that it does not participate in a circuit with its own neuronal dynamics, as the authors do here, then the forward simulation and reverse inference are consistent with each other. However, this rarely the case, changing synaptic time constants and amplitudes will change the local recurrent neural dynamics, this is well known. In fact, the mechanism through which propofol acts is likely that the local and global neural dynamics are altered due to a change in the GABA synaptic properties.

I would emphasize that this is a point that should be discussed as a limitation of both the model, and the interpretation of the EEG data (and any real data), in the discussion section. I don't think more extensive modeling is required from the authors, but the fact that these different factors mix together in their causal impact on the final measured EEG is exactly the reason reverse inference is so difficult should be communicated, and extending the modeling study to consider this would be an exciting future direction.

Response 2: We completely agree with the reviewer and indeed plan to pursue this research direction in future work. As already stated in our Discussion, propofol's action on GABAR currents likely decreases network criticality which will lead to its own changes in the EEG signal. We have taken the advice of the reviewer and emphasized this phenomenon in the Discussion as a limitation to detrending real EEG data (lines 358-365).

3. As per the reviewer response, I think showing the tau change for all the channels would be great as a supplemental figure (Response 4 under reviewer 3 in the response letter)

Response 3: We now show the changes in tau associated with all channels in Supplementary Figure S7.